# Biomolecular Composition and Revenue Explained by Interactions between Extrinsic Factors and Endogenous Rhythms of *Saccharina latissima*

**DOI:** 10.3390/md17020107

**Published:** 2019-02-10

**Authors:** Xueqian Zhang, Marianne Thomsen

**Affiliations:** Department of Environmental Science, Aarhus University, Frederiksborgvej 399, DK-4000 Roskilde, Denmark; xqzh@envs.au.dk

**Keywords:** seasonal and spatial variations, physiological and ecological characteristics, ecosystem services, nature-based solutions, water quality restoration, blue growth

## Abstract

This review provides a systematic overview of the spatial and temporal variations in the content of biomolecular constituents of *Saccharina latissima* on the basis of 34 currently available scientific studies containing primary measurements. We demonstrate the potential revenue of seaweed production and biorefinery systems by compiling a product portfolio of high-value extract products. An investigation into the endogenous rhythms and extrinsic factors that impact the biomolecular composition of *S. latissima* is presented, and key performance factors for optimizing seaweed production are identified. Besides the provisioning ecosystem service, we highlight the contribution of green-engineered seaweed production systems to the mitigation of the ongoing and historical anthropogenic disturbances of the climate balance and nutrient flows. We conclude that there are risks of mismanagement, and we stress the importance and necessity of creating an adaptive ecosystem-based management framework within a triple-helix partnership for balancing the utilization of ecosystem services and long-term resilience of aquatic environment.

## 1. Introduction

There is increasing interest in using seaweed for its potential use as a feedstock in various applications [1]. Asia has a long history of cultivating seaweed for human consumption, as well as other uses, such as the production of soil improver, feed supplements, and hydrocolloids [2]. In the Western world, the interest in seaweed cultivation originates from its potential role as a third-generation feedstock for biofuel production. However, more recently, the focus of research and development (e.g., in the research and innovation projects MacroAlgae Biorefinery for Value-Added Products (https://mab4.org/), Macro Cascade (https://www.macrocascade.eu/), SeaRefinery (https://searefinery.eu/), and FucoSan (https://www.fucosan.eu/en)) has shifted toward the extraction of bioactive molecules in seaweeds with unique functional properties. Bioactive compounds of interest include amino acids, *n*-3 unsaturated fatty acids, and species-specific compounds, such as fucoidans and fucoxanthin in kelp species (Phaeophyceae) and ulvan in Chlorophyceae.

*Saccharina latissima* is one of the native kelp species found in the North Atlantic and North Pacific (sketch shown in Figure 1; general information about *S. latissima* presented in Table 1) and has received growing interest for large-scale offshore cultivation due to its potential for high biomass yield and its richness in valuable biochemical compounds. Common *S. latissima* production systems, including long-line [3] and integrated multitrophic aquaculture (IMTA) [4], are mostly designed using processes (nursing, deployment, and harvesting) that mimic the natural reproduction cycle of *S. latissima* and parameters that replicate the favorable environmental conditions identified for wild stocks [5].

Many research studies have demonstrated its valorization potential as a feedstock via biorefining. The output products from a *S. latissima* biorefinery range from bioethanol [6], succinic acid [7], biogas and phenolic compounds [8], pinene [9], bioethanol, liquid fertilizer, and protein-rich fish feed ingredients [10] to biomaterials for medical devices [11].

However, there exists one major bottleneck in the development of large-scale *S. latissima* farming and industrial applications of its valuable constituents: its ever-changing biomolecular profile. For complex and heterogeneous biomolecule mixtures, such as fucoidans, substantial variation in chemical structure and composition constitutes a barrier to their market entry and the application of their known bioactivities as medicinal products, e.g., as an inhibitor of breast cancer [12]. Due to the challenges in reproducing them, medicinal products containing fucoidans can hardly fulfill the strict pharmaceutical quality requirements for medicinal products that are essential for product efficacy and safety [13]. Therefore, despite intense research during the last few decades, fucoidans are currently only utilized as ingredients in food supplements and cosmetics, while a medicinal product containing fucoidans is yet to be approved [13]. To improve biomass utilization and valorization, it is therefore necessary to enhance the current knowledge of the endogenous rhythms and extrinsic natural and anthropogenic factors that regulate and alter the biomolecular composition of *S. latissima*.

Besides the provision of biomass and bioactive substances, offshore seaweed cultivation provides significant ecosystem services, such as water quality restoration and climate change mitigation. Cultured *S. latissima* delivers the supporting service of nutrient cycling owing to its nitrogen (N) and phosphorous (P) absorption during the growth phase, and this helps remediate eutrophication of aquatic systems. Moreover, *S. latissima* takes up inorganic carbon (C) and forms cellular biomolecules in light-independent carbon fixation processes with the help of carboxylating enzymes, thereby contributing to climate change mitigation [14]. These services increase the ecological importance of seaweed production and, more significantly, lead to the creation of a resilient seaweed-based bioeconomy that is characterized by the circular nature of the global biogeochemical cycle [15].

The present review paper provides a systematic overview of the state-of-the-art knowledge within the existing research literature on spatial and temporal variations in the biomolecular composition of *S. latissima*. Secondly, to assimilate the observed variations in the biomolecular composition of *S. latissima*, this study reviews the physiological characteristics underlying the response of *S. latissima* to various environmental factors. Thirdly, the potential revenues that are associated with the different biomolecular constituents of *S. latissima* biomass as a function of harvest time and cultivation site are analyzed, and the key aspects of seaweed production practices that are designed to optimize revenues from high-value extractables are discussed. Lastly, we propose the creation of an adaptive ecosystem-based management framework for ensuring the sustainable development of a macroalgal bioeconomy that contributes to the restoration of the balance of Earth’s biogeochemical cycles. This latter point is addressed by highlighting the multifunctional aspects of offshore seaweed production systems as an instrument for circular nutrient management, thus delivering a high-value feedstock for biobased industries while contributing to the mitigation of the ongoing eutrophication of aquatic systems, ocean acidification, and climate change.

## 2. Methods and Data

### 2.1. Literature Review of the Biomolecular Composition of *S. latissima*

Data on the biomolecular composition of *S. latissima* were obtained from the scientific publications by searching the databases of the Web of Science and Google Scholar for the words TS= (composition OR seasonal variations OR spatial variations OR protein OR fucoid?n OR laminar?n OR alginate OR alginic acids) AND (Saccharina latissima OR Laminaria saccharina OR sugar kelp). We used the Boolean operator “OR” and the question mark (?) to broaden the retrieval. This ensures that all records that refer to any of the search terms by using either the exact words or equivalents (e.g., synonyms and words with alternate spellings) are included. The operator “AND” was used to narrow the search results to the references that are relevant to *S. latissima*.

Furthermore, we carried out a screening of the retrieved records according to PRISMA guidelines to limit the scope to primary research articles containing the composition data of *S. latissima* samples, which were collected from the wild stocks or offshore cultivation systems. In the first step of the screening, we removed duplicates, Ph.D. theses, and MSc theses. Secondly, studies on the *S. latissima* cultivated in land-based systems were excluded. Ultimately, 34 articles were selected for the review of the biomolecular composition of *S. latissima* (detailed description provided in Appendix B, Figure A1).

Data and information on the biomolecular composition of *S. latissima* were collected from tables and figures, and the background information of each sample was obtained from each study’s Material and Methods section and Appendix A. A complete and systematic overview of the collected primary data and information can be found in the review database on location-specific environmental parameters and biomolecular composition (Appendix A). Data pretreatment and analysis, including a graphical representation of composition data, were carried out in Microsoft Excel.

### 2.2. Literature Search of Physiological Characteristics of *S. latissima*

A second literature search conducted on the Web of Science and Google Scholar platform was focused on the physiological characteristics of *S. latissima* that underlie its responses to extrinsic stresses. The primary search term applied was “Saccharina latissima OR Laminaria saccharina OR sugar kelp”. This term was combined with the terms “endogenous rhythms”, “irradiance OR light”, “temperature”, “salinity OR osmosis stress*”, “exposure OR water motion”, and “nutrient”, among others, by using the Boolean operator “AND”.

### 2.3. Revenue Analysis of Biorefinery Output Products

A simplified revenue analysis was conducted according to a valorization method that was previously applied to biowaste [16]. We considered the most simple and immediate scenarios of Phase I biorefineries, which use only one feedstock and only have a single major product at a time [17].

The associated revenues of the extract products were quantified according to the biomolecular composition of *S. latissima*. In the calculation, we took into consideration the seasonal and spatial variations in the tissue contents of specific biomolecules (*B*), as well as the seasonality of dry biomass content (DB). Market prices of the output products were collected from research articles, reports by academic institutions and industrial associations, and personal correspondence with the sales departments of stores. The prices (*P*) applied in the calculations are the average values of the crude products’ prices (Table 2). The potential revenue (*R*) expressed in units of kg fresh weight seaweed biomass was calculated according to Equation (Equation 1):(1)R$kgfreshbiomass=P$kgbiomolecule×Bkgbiomoleculekgdrybiomass×DBkgdrybiomasskgfreshbiomass

## 3. Results

### 3.1. Seasonal and Spatial Variations in Biomolecular Composition of *S. latissima*

#### 3.1.1. Dry Biomass Content on a Fresh Biomass Basis

The dry biomass content of *S. latissima* changes seasonally, with an overall trend that rises over the spring, peaks in the summer, and is at its lowest in the winter (plot shown in Figure 2). The reported values of dry biomass content on a fresh biomass (FB) basis range from 5.3%–36%.

#### 3.1.2. Content of Biomolecules on a Dry Biomass Basis

Ash, representing minerals, and metal ions constitute the greatest portion of the dry biomass of *S. latissima*. Carbohydrates are also major constituents of dry *S. latissima* biomass (a summary of their functions, fundamental subunits, and subunits’ formula are presented in Table 3). In addition, other macromolecules (such as proteins and lipids), together with other potential high-value compound groups (such as phenolic compounds) are commonly found in *S. latissima* in smaller fractions.

##### Ash

The ash in *S. latissima* consists of inorganic salts and cations, which are assimilated from the local aqueous environment and partially retained in the intracellular fluid inside cell vacuoles and partially absorbed at the outer cell surface of the algae. The cations are part of the organic molecular ion-chelating structure in the extracellular matrix, where they are bound with structural components, such as alginic acids and fucoidans.

Figure 3 presents the ash content reported in the literature, as well as measurements of minerals [34] and the sum of the four most abundant metal ions in *S. latissima*, i.e., K+, Na+, Ca${}^{2+}$, and Mg${}^{2+}$ [35,36] (measurements of the content of individual metal ions can be found in the Appendix A).

As shown in Figure 3, the seasonal variation in the ash content of *S. latissima* is pronounced. The highest reported ash content reaches up to 47% DB, while the lowest reported value is 14% DB. There is a clear and common temporal trend of ash content across all long-term monitoring studies at the sites Kattegat, Hjarnø, Shuna Island, Eilean Coltair, and Clachan Sound (Figure 3): i.e., the maxima of the ash content appear in the winter months from January–March, and the minima occur around the summer from July–September. In contrast, short-term datasets based on samples collected from the sites Færker Vig, Taraskjæret, and Frøya Island show a general tendency of ash content to increase over the summer.

##### Alginate 

Alginate is an important cell wall component in all brown seaweed [37]. It is the salt form of alginic acid, an anionic polysaccharide composed of D-mannuronic acid (M) and L-guluronic acid (G) residues connected via 1,4-glycosidic linkages [38]. M and G are present in different proportions and form linear chains of varying proportions of homopolymeric (MM or GG) or heteropolymeric (MG or GM) blocks [39]. The molecular weight of alginate generally ranges between 500 and 1000 kDa [40].

Figure 4 shows that the alginate content in *S. latissima* varies considerably with the seasons in a wide range (4.1–34.5% DB). The peak values commonly occur in late spring (from April–June) irrespective of sample origins, while the accumulation and declining patterns of alginate content differ slightly between wild and cultivated *S. latissima* samples. The data on wild *S. latissima* samples reveal a common gradual increase in alginate content over a long period, from late autumn to the following spring. Contrary to all others, samples from Clachan Sound and the Barents Sea tend to have higher alginate content in September.

##### Mannitol

Mannitol is a water-soluble non-reducing sugar alcohol present in many species of brown algae, especially in Laminariales and Ecklonia [1]. In *S. latissima*, mannitol is a primary photosynthesis product derived directly from the photoassimilate fructose-6-phosphate (F6P) via the action of mannitol-1-phosphate dehydrogenase and mannitol-1-phosphatase [41].

The mannitol content in *S. latissima* is subject to wide seasonal fluctuations across all sites, varying between 0.5% and 24% DB, as presented in Figure 5. Overall, there is a dominating upward trend in mannitol content over the winter months, and after reaching its peak in the period from April–June, mannitol content starts declining and then reaches its lowest value in September. Generally, the mannitol content in samples from natural stocks rises and falls more moderately compared with that in cultivated *S. latissima* samples, and it stays at the maximum level for a longer period of time with slight fluctuations (e.g., datasets from Kattegat, Shuna Island, Eilean Coltair, and Clachan Sound in Figure 5). For cultivated *S. latissima*, late deployment (e.g., May for the sample from Fornaes; background information of all datasets is provided in the Appendix A) is likely to cause a delay in the peak of mannitol content (i.e., September and May in the first and second cultivation year, respectively) and a lower maximum concentration (6% DB and 5.1% DB in the first and second cultivation year, respectively).

##### Fucoidans 

Fucoidans are a group of polysaccharides primarily composed of sulfated L-fucose with other monosaccharides (such as uronic acids, galactose, xylose, rhamnose, and glucose) and sugar acids (glucuronic acid) in small proportions [42]. They are unique matrix polysaccharides widely found in the cell walls and intercellular spaces of brown seaweed, but not in other algae or higher plants [1].

Extensive research studies in the past few years have investigated the molecular congeners of fucoidans isolated from *S. latissima* and other brown seaweed species, and the chemical structure and biomedical properties of their fucoidan extracts have been elucidated [43]. However, even crude fucoidans isolated from the same brown algae species mostly consist of a mixture of structurally-distinct polymers, and the composition varies; therefore, so far, no generalizations have been made for the structural characteristics of fucoidans in *S. latissima* [13,44,45,46].

For the same reasons, it has been difficult to reach a general consensus on the terminology for fucoidans. Fucose content is commonly adopted as the reference for fucoidan content on the basis of the significant linear relationship found between the content of crude fucoidans and fucose content. Moreover, a few studies have used such terms as “fucose-containing sulfated polysaccharides” [13,47] and “sulphated fucans” [48,49] to describe fucoidans. Due to the lack of a standardized quantification method, fucoidan extracts reported by different studies vary considerably in their degree of purity, composition, and bioactivity.

The plot in Figure 6 combines all measurements related to fucoidans from the literature, including the measurements of fucoidans, fucose, combined L-fucose, and crude sulfated polysaccharides (CSP). The reported fucoidan content in *S. latissima* ranges from 1.3%–8.8% DB. Overall, there are no pronounced seasonal patterns except for the sample from Kattegat, whose fucoidan content varies with an amplitude factor of 2.6 (reaching the maximum (6.2% DB) and the minimum (2.4% DB) in August and January, respectively). The fucoidan content displayed in Figure 6 was quantified by measuring total fucose content; most of the datasets in the figure reveal a seemingly constant level of fucoidan content, indicating that this measurement approach quantifies only the backbone structure of fucoidans. On the other hand, the measurement methodology applied to the Kattegat samples in the study by Bruhn et al. (2017) [46] also accounts for the monomers constituting the branching structure of fucoidans. Thus, the measurements in the study by Bruhn et al. (2017) [46] are most likely to reflect the development of fucoidans’ branching structure, whose complexity level increases with higher photosynthetic activity in the spring–summer period.

##### Laminarin 

Laminarin is a principal and unique carbohydrate reserve substance in *S. latissima* that is located intracellularly in vacuoles [50]. It is found in Laminariales and, to a lesser extent, in *Ascophyllum*, *Fucus*, and *Undaria* species [1]. Laminarin is basically a class of low-molecular-weight linear polysaccharides and primarily composed of (1,3)-β-d-glucan with some β-(1,6)-linked branching in the main chain, although β-(1,6) interchain links may occasionally occur [40]. There are two types of laminarin chains, M and G, which differ in their reducing ends. M chains end with a mannitol residue, whereas G chains end with a glucose residue, and they are present in about a 3:1 ratio [51]. Laminarin’s molecular weight is approximately 5 kDa depending on the degree of polymerization (usually 20–25), and its solubility increases with the increasing degree of branching [40].

Laminarin is the most important glucose-containing polysaccharide in *S. latissima* [52]; the other is cellulose, whose content in *S. latissima* is, however, fairly low; thus, glucose derived from cellulose is found in negligible amounts [53]. As such, glucose content is often taken as a proxy for laminarin content, and variations in glucose content and laminarin content are often discussed together within the scientific literature. The present study verifies that they do share the same pattern, which can be observed in Figure 7.

Laminarin content shows substantial variation across different sampling months and locations, ranging from 0%–39% DB. Overall trends in the data show a tendency toward a late summer peak for laminarin in the wild *S. latissima* samples and a slight tendency toward an early summer peak in cultivated *S. latissima*, while a low and constant level of laminarin content is observed in the samples collected from eutrophic and sheltered waters, e.g., Færker Vig. In general, laminarin follows the same seasonal pattern as mannitol for the wild stocks.

##### Protein 

The protein fraction of brown algae is generally lower than that of green and red algae species [2,47], as well as that of land-based plants, such as soybeans, where protein makes up to 35.8% of the dry biomass [54].

As shown in Figure 8, the protein content in *S. latissima* varies between 1.4% and 24.2% DB. Both wild and cultivated *S. latissima* samples display an overall tendency toward reaching their maximum protein content in early spring (February and March) and their lowest level in summer (from July–September); for the late-deployed (May) Hjarnø samples, a delayed peak in protein content is observed (November), and maxima are reduced to only 10%.

Regarding the composition of *S. latissima* protein, the most abundant amino acids are glutamic acid, aspartic acid, and alanine, accounting for 30–50% of the total amino acids (TAA) [34,55,56,57]. This is in accordance with the observations by Dawczynski et al. (2007) [58] for *Laminariales* species, while the amino acids with the highest content in *S. latissima* protein reported by Mai et al. (1994) [59] are arginine, leucine, and lysine.

##### Lipids 

Lipids cover a broad spectrum of naturally occurring molecules, such as waxes, sterols, fat-soluble vitamins, and phospholipids [1], among others. Fatty acids and carotenoids in brown algae have received the most attention, primarily due to their health-promoting activities.

There are only three datasets available in the literature revealing the changing lipid content in *S. latissima* over the seasons, as shown in Figure 9. Monocultured and IMTA *S. latissima* from Hjarnø show similar trends; i.e., lipid content reaches a peak of around 3.4% DB in November and then starts to decline until reduced to the minimum levels (slightly below 1% DB) in July. A different pattern is observed in wild samples collected from Amorosa for total fatty acids (TFA) content, which remains at low levels (around 0.2% DB) during the period from July–December, after which it rises and reaches 1% DB in May. Lipid content reported by other studies (including single and short-term monitoring measurements) generally ranges between 0.1% and 1.3% DB, while the lipid content of specimens collected from Fink Cove in April is considerably higher at 5.5% DB.

##### Phenolics 

Phenolic compounds consist of one or more aromatic rings with one or more hydroxyl groups [60], ranging from low-molecular-weight and single aromatic-ringed compounds (e.g., simple phenols) to large and complex tannins and derived polyphenols [61]. There exists a high level of heterogeneity in the broad classifications of phenolic compounds used in the literature, as well as in the isolation and characterization methodologies [62]. Apart from terms such as “total phenolic compounds (TPC)” and “polyphenols”, the term “phlorotannins”, which describes oligomers of phloroglucinol [63], is often used to refer to phenolic compounds in kelps. Earlier reported values of phlorotannins in kelps range between 0.6% and 5.3% DB [64,65].

Figure 10 plots combined TPC measurements (TPC expressed in % DB, TPC expressed in gallic acid equivalents (GAE) per kg DB, and phlorotannin content) reported in the literature for *S. latissima*. Overall, phenolics in *S. latissima* exhibit a fairly constant concentration level across most samples and seasons. There is a significantly higher content of phenolics (5.1% DB) for the specimens from Hjarnø that were cultivated over two growing seasons, and this is in agreement with the finding for *Laminaria hyperboren*, another common kelp species, whose older algal tissue generally contains more phenolic compounds compared with younger tissue [65].

### 3.2. Endogenous Rhythms and Extrinsic Factors

This section presents an overview of *S. latissima*’s key physiological and ecological characteristics that underlie its responses to varying environmental conditions and accordingly affect its biomolecular composition (observations of compositional variation are presented in Section 3.1).

Section 3.2.1 briefly describes the functional form of *S. latissima* and the endogenous rhythms of C and N metabolism throughout its life history. Section 3.2.2 summarizes the state-of-the-art knowledge of the influences of extrinsic factors on the bioactivities (growth, development, and metabolism) of *S. latissima*, as well as the roles of biomolecules in coping with different environmental stresses. To provide a snapshot of this section, we visualize the potential interactions between key biomolecules, bioactivities, and extrinsic factors of *S. latissima* in Figure 11.

#### 3.2.1. Endogenous Rhythms under Natural Conditions

##### Functional Form of *S. latissima*


*S. latissima* has an advantageous morphology for light harvesting and nutrient uptake, i.e., thick blades. Internally, *S. latissima* is differentiated, heavily corticated, and thick-walled. As a multicellular eukaryote, *S. latissima* has tissue systems made of various cell types that possess specialized morphological and physiological features and carry out specific functions.

Many studies provide scientific evidence of intra-thallus variations in terms of cell construction, as well as biomolecular content and structure. For example, a blade of *S. latissima* contains a higher density of thylakoid and mitochondrial cristae per unit volume [66]. Higher contents of TFA and pigments, which play an important role in the structural integrity of the thylakoid membrane of the chloroplast, were found in blades compared with stipes [68], and the TFA in blades contained more *n*-3 polyunsaturated fatty acids (PUFA) [69]. In another instance, the stipe—the perennial bit of *S. latissima* thallus—was shown to possess higher amounts of phenolic compounds than the blade (approximately 40% vs. 3.5% DB) [64]. This was attributed to phenolic compounds acting as secondary metabolites in the life history strategy of *S. latissima*. Thus, older algal tissue is likely to contain more phenolic compounds [65].

Even for the same organ, there is a clear delineation between different tissue types along the cross-section. In general, pigmented cells tend to be located around the periphery of thalli, and they are small, polyhedric, and contain numerous chloroplasts, while medullary cells are large, round, and have few or no chloroplasts [66,70].

Such dispositions facilitate and improve light absorption and the passage of inorganic carbon (in the form of CO${}_{2}$ and HCO${}_{3}^{-}$) for photosynthesis (linkage a in Figure 11). Primary photosynthetic products are utilized in various ways to serve different metabolic needs. Apart from their immediate use in such processes as respiration, primary photosynthetic products are utilized in conjunction with nitrogen (provided that ambient N concentrations are high) to produce amino acids and proteins for growth. Alternatively, these products are utilized to form large storage and structural polysaccharides (Table 3).

The polysaccharides of *S. latissima* display a high degree of structural heterogeneity [53]: from the primary structure determined by the types of monosaccharides, to the secondary structure decided by the different glycosidic bonds linking monomers, and to the tertiary structure that defines how polysaccharides may be linked to each other [71]. After polymerization, further complexity is added to algal polysaccharides via the conversion of residue units, the addition of sulfate and methyl groups, the formation of anhydride bridges, etc. One example is the sequential biosynthesis of sulfated fucans, which are likely to be polymerized as neutral polysaccharides and then sulfated by specific sulfotransferases [13,49].

Figure 11 presents one of the possible scenarios of algal cell wall construction according to the state-of-the-art knowledge obtained from the most recent scientific literature [43,49,66,67]. In the algal cell wall, there is no lignin, and cellulose accounts for only a small proportion [8]. The algal cell wall structure is mainly based on an alginate network and fucoidans, which fill the space between cellulose microfibrils [49]. As the primary component of the algal cell wall, gel-forming alginate plays an important role in cell wall organization and affects its rigidity (linkage b in Figure 11). Alginic acids containing a higher proportion of G units have a substantially higher affinity for cations (Ca${}^{2+}$) compared with the ones rich in M units; therefore, alginate (the salt form of alginic acids, introduced in Point 2 of Section 3.1.2) made of the former increases the stiffness of the cell wall and is predominant in the stipe, while alginate made of the latter adds elasticity to the cell wall and is more typical of the blade [66]. Fucoidans appear to confer adaptive advantages to algal cells, and a few studies have provided experimental support for the assumption that fucoidans are involved in ionic and osmotic regulation [13,48].

Phenolic compounds are commonly regarded as having primary metabolic roles in wound healing [72] and constitute an algal defense against stress conditions and herbivores [63,73] (linkage c in Figure 11). In the brown algae cell wall matrix, phenolic compounds are linked to alginate, while proteins are tightly associated with fucoidans, cellulose, and phenolic compounds [67]. This cross-linking between integral components of the algal cell wall is key to regulating the strength of cell walls and maintaining cellular integrity [74].

##### Internal C and N Reserves of *S. latissima*


*S. latissima* has a seasonal cycle of reproduction and growth by nature. The winter months and the first half of the year are usually characterized by blade growth, which then slows during the summer months [5,75]. In autumn, the old blade is lost by natural shedding or necrosis [70], concomitant with new growth in the intercalary meristem. Figure 12 sketches one life cycle of *S. latissima*. First, new growth (gametophyte stage) of *S. latissima* starts with the release of spores from mature sorus tissue, which is at the bottom of the frond and directly above the stipe (sketch presented in Figure 1). During the meiosis process, haploid meiospores are released, settle, shed flagella, and develop into microscopic male and female gametophytes. At the end of germination, eggs extruded by female gametophytes attract the flagellated spermatozoids from nearby male gametophytes. Then, diploid zygotes growing from the fertile eggs divide longitudinally and develop into small juvenile sporophytes.

The above life cycle rhythm of *S. latissima*, together with natural cycles of light and nutrient availability in the environment, determines the biomolecular composition of *S. latissima* by regulating the respiration, photosynthesis, and growth processes during which biomolecules are synthesized, degraded, accumulated, or internally translocated. Hereafter, particular emphasis is put on the internal C and N reserves in *S. latissima*, because C is the most fundamental element of seaweed biomass and plays an essential role in forming cellular molecules, and N is an element of major metabolic significance and is most frequently limiting to seaweed growth in marine environments.

*S. latissima* tends to build up an internal soluble nitrogen reserve in the winter months when sufficient ambient dissolved nitrogen (in the form of NH${}_{4}^{+}$, NO${}_{2}^{-}$, NO${}_{3}^{-}$, urea, and amino acids) facilitates nitrogen uptake while low levels of light and temperatures limit growth [76]. *S. latissima* grows by utilizing mostly exogenous nitrogen, and this practice lasts until late spring and summer of the following year when phytoplankton exhausts the exogenous nitrogen in the aquatic system. To overcome exogenous nitrogen shortage, *S. latissima* starts utilizing the previously-stored internal nitrogen reserve to initiate sporophyte growth. This internal nitrogen stock declines and becomes depleted around the summer, e.g., July [77].

In *S. latissima*, there seems to exist a biochemical route that connects mannitol and laminarin (linkage d in Figure 11), as hypothesized for the brown algae species *Ectocarpus siliculosus* [49] and *Saccharina japonica* [78]. In the summer months, when abundant irradiance prompts photosynthesis and nitrogen is limiting under natural conditions, a considerable amount of mannitol is produced and incorporated into laminarin. The laminarin reserve acts as long-term carbon storage and starts being exploited by *S. latissima* to compensate for the carbon loss in the respiration process in late autumn and winter as light conditions become worse and, accordingly, photosynthetic rate declines [79,80].

The stored C and N reserves not only can serve the nutrient demands of local tissues, but also can be mobilized and translocated to other parts of the thallus. Intra-thallus translocations of C and N follow a “source-to-sink” pattern and only occur when there concomitantly exists a nitrogen or carbon deficit in the basal meristem (the sink) and a surplus of photosynthate in the mature apical blade (the source). In this biological transportation network, mannitol and amino acids (chiefly alanine, glutamic acid, and aspartic acid), together with inorganic ions, are moved from the apical blade to basal meristem through trumpet-shaped sieve elements [81] and become rapidly metabolized and incorporated into polysaccharides and proteins in the meristematic tissues [82]. Mannitol and amino acids each constitute about half of the exported carbon [66]. For the meristematic nitrogen demand, translocated amino acids account for 70%, while the remaining 30% comes from the assimilation of exogenous sources [83]. However, intra-thallus translocation may sometimes be ineffective due to abscission or necrosis, therefore leading to a dramatic drop in the net C or N content in the biomass [70,77].

#### 3.2.2. Extrinsic Factors and Potential Stresses

Although self-sustained, the rhythms of *S. latissima* can be adjusted (or entrained) to the local environment by external cues. Abiotic environmental factors, including irradiance, temperature, water motion, salinity, and availability of nutrients, together with biotic factors, can impact bioactivities and thus the biomolecular composition of *S. latissima*. Extreme conditions, in which environmental parameters are far from the optimum (listed in Table 4), can exert pressures on *S. latissima*, impairing its reproduction and, moreover, threatening its survival.

##### Irradiance 

The availability of sunlight for macroalgae in a marine environment is naturally limited compared with land-based plants, and this is especially the case for seaweed in deep water or the turbid zone. Light at longer wavelengths with lower energy is often absorbed or scattered near the water surface; thus, only a restricted amount of light at shorter wavelengths can penetrate the deeper zone. Moreover, not all light can be utilized by *S. latissima* due to the characteristic absorption spectra of different photosynthetic pigments. For instance, peak light absorption lies at 470 nm for carotenoids and 630 nm for phycoerythrin, while chlorophyll a has two peaks, with the primary one at 440 nm and a smaller peak at 670 nm [90].

The adult sporophyte of *S. latissima* has a high tolerance for the underwater light climate, as indicated by the dense stocks in the high-arctic fjord, where ambient irradiance at a 10 m depth varies greatly from as high as 200 μmol m${}^{-2}$ s${}^{-1}$ during the open-water period to 3–6 μmol m${}^{-2}$ s${}^{-1}$ under ice cover [91].

In cases of light limitation, *S. latissima* can adopt a set of mechanisms, such as optimizing its light-harvesting process and lowering its respiration rate with the help of low temperature, to maintain the carbon balance and ensure growth and survival. In the specimens collected from turbid water, Gerard (1988) [92] found higher concentrations of pigments and N, and this phenomenon was ascribed to the increased production of photosynthetic pigment–protein complexes and enzymes, which can improve light absorption and facilitate photosynthesis (linkage e in Figure 11). Arctic *S. latissima* in Young Sound compensates for the low availability of light by maintaining old blades that are able to contribute to inorganic carbon acquisition [91]. In another instance, Barbosa et al. (2017) [69] observed a tendency toward higher levels of TFA in *S. latissima* at deeper locations along the water column. This could be explained as a life strategy response to reduced light conditions; i.e., storage of fatty acids as energy source supplying adenosine triphosphate (ATP) for carbon fixation.

Photoprotection and photoinactivation are two co-occurring mechanisms that allow *S. latissima* to recover from photoinhibition when it confronts excessive irradiance. Photoprotection is a reversible process with fast reaction kinetics that regulates the quantum yield of photosynthesis by increasing thermal energy dissipation; photoinactivation is a slow recovery process that works to repair damaged reaction-center proteins, and it takes place only slowly in dim light [93]. Protective pigments (e.g., xanthophylls) are involved in the photoprotective mechanism, and their conversion is possibly controlled by carotenoids [94]. Phenolic compounds, which are among the most powerful antioxidant biomolecules in *S. latissima* [73], are also involved and act as a buffer against highly reactive oxygen molecules originating from excess light absorbed by chlorophylls [8].

##### Temperature 

*S. latissima* is a cold water-temperature species with a broad temperature optimum in the 10–15 ${}^{\circ }$C range [85]. As indicated by its survival in high-arctic areas [90,91], *S. latissima* possesses growth strategies to adapt to low-temperature environments. One commonly identified adaptation method is the desaturation of fatty acids, which are central building blocks and important constituents of cell membranes that play a key role in the growth and development of organisms [95]. Like mammals, brown algae in cold waters change their metabolism to switch to the pathway of utilizing saturated fatty acids (SFA) for building up PUFA, thus maintaining cell membrane fluidity, as well as prevent low-temperature photoinhibition [96,97] (linkage f in Figure 11). Besides increased total unsaturation, there appears to be a bigger fraction of *n*-3 PUFA in TFA in cold waters [98] and in cold seasons [55].

Moderate heat can stimulate the growth of *S. latissima*. A large-scale comparison study conducted in a Danish embayment demonstrated that warming increased the level of peak growth and seemingly advanced and expanded the growth period of *S. latissima* [77]. Experiments investigating the growth and photosynthetic performance of Arctic populations in response to increased temperature and pCO${}_{2}$ levels in the marine environment reveal ecotypic differences, and arctic populations of *S. latissima* seem able to take advantage of the impending scenario of ocean warming and acidification; this is in stark contrast to its counterparts in the Atlantic Ocean [99].

High-temperature regimes in the summer impact sporophyte growth adversely, the degree to which depends on the duration and frequency of hot summer days, as well as the amplitude of seasonal temperature changes [100]. Bolton and Lüning (1982) [85] investigated the heat threshold of *S. latissima* and found that at 20 ${}^{\circ }$C, the growth was reduced by 50–70%, and complete disintegration occurred after 7 days at 23 ${}^{\circ }$C. In another test, *S. latissima* suffered 100% mortality after 3 days at 24 ${}^{\circ }$C [92]. Excessive heat has been considered the most plausible driver of the large-scale loss of kelp forest along the southwest coast of Norway between the late 1990s and early 2000s [75]. It was found that the acquisition of heat tolerance is affected by growth temperatures. In experiments by Gerard (1988) [92] in the extreme southern limit of the geographic range, specimens grown at 0–24 ${}^{\circ }$C survived for at least 6 weeks at 20 ${}^{\circ }$C, while specimens grown at 0–18 ${}^{\circ }$C suffered 100% mortality during the third week.

##### Salinity 

As a marine species, *S. latissima* is able to osmoregulate, but this ability is insufficient to tolerate a wide range of salinities. The effect of salinity on the biogeography of *S. latissima* is exemplified by the phenomenon of the “downward process” [101]: namely, *S. latissima* tends to move downward to deeper water columns in response to lowered salinity near the sea surface and the resultant increase in competitive interactions with euryhaline species [102]. Further, salinity affects physiology and even the genetic structure of local populations of *S. latissima*. Low salinity was determined to be one contributing factor to the relatively low growth rates of *S. latissima* populations in a Danish embayment [77]. In this study, 10 per mille was identified as a critical point, below which the photosynthetic CO${}_{2}$ uptake of *S. latissima* was reduced irreversibly [103]. The ecotypes along the salinity gradient area of Danish waters, which stretch from the fully marine North Sea (around 25 per mille) to the brackish Baltic Sea (around 10 per mille), show clear genetic divergence and lower genetic biodiversity in comparison with their marine counterparts [102].

On the cellular level, the most important biochemical effects of salinity are changes to water potential and electrochemical potential, both of which promote the flow of water molecules and ions, as well as the interconversion of monomeric and polymeric biomolecules [66]. Inorganic ions, such as K${}^{+}$, Na${}^{+}$, and Cl${}^{-}$, whose concentrations can be rapidly adjusted, are involved in the short-term osmotic acclimation in vacuoles (linkage g in Figure 11) [104]. In cytoplasm and organelles, including mitochondria and chloroplasts, water potential is regulated by mannitol, which acts as a compatible solute and is the main organic osmolyte in brown algae [105]. Fucoidans and iodine may also hold an osmotic function. Fucoidan content and degree of sulfation (DS) in a comparison study concerning *S. latissima* specimens collected from Kiel Fjord (around 15 per mille) and the Faroe Islands (around 35 per mille) appeared to be positively correlated with the salinity levels of the specimens’ living environments. The tissue iodine content of the brown algae *Laminaria digitata* was found to increase in response to hyposaline conditions, while net I${}^{-}$ release rates increased in response to declining salinity levels in an ambient environment. Moreover, the results indicated that, under hypersaline conditions, I${}^{-}$ may have substituted the role of mannitol [106].

##### Nutrients 

*S. latissima* requires a sufficient supply of nutrients for healthy growth. Essential elements (C, H, O, N, P, Mg, Cu, Mn, and etc.) play irreplaceable metabolic roles, and their availability in an ambient environment determines the biomolecular composition, growth, development, and distribution of *S. latissima* [66,77,80,107].

Particular emphasis in this section is put on N and P. Among the various nutrients required by *S. latissima*, these two elements are most closely associated with marine ecosystem health in terms of aquatic eutrophication. Moreover, N and P availability in aquatic systems are most likely to be affected by local anthropogenic activities, e.g., runoff from agricultural land leading to increased N and P levels and reduced N and P concentrations due to eutrophication mitigation actions.

There appears to be an approximately positive linear relationship between *S. latissima* growth and ambient N concentration in the winter [80] and spring [108]. With increasing ambient nitrate concentration, chlorophyll content and photosynthetic capacities of *S. latissima* were found to increase [109], and this is presumably due to the increased formation of pigment–protein complexes in the chloroplast. In contrast, the summer growth rate is more likely to positively correlate with internal reserves of N that are accumulated during the winter, particularly soluble organic nitrogen reserves [108].

In the algal cell, P is characterized by its decisive roles in three essential structures: the cell membrane (e.g., phospholipids); the energy transfer system (e.g., ATP); and the storage and retrieval system for genetic information (i.e., nucleic acids) [110]. A deficiency in P (principally in the form of PO${}_{4}^{3-}$) in ambient waters can therefore limit algae growth and the completion of its reproductive cycle (linkage h in Figure 11). Although P is commonly known as a non-limiting factor in marine waters, there exist some significant exceptions, e.g., Limfjorden, Denmark [14,111], and the Baltic Sea [112], where low levels of P limited *S. latissima* growth [102] and possibly led to lowered mannitol concentrations [52].

##### Water Motion 

Unique physiological characteristics of *S. latissima* allow it to adapt to the ever-changing hydrodynamic conditions in the offshore environment. The holdfast of *S. latissima* (sketch shown in Figure 1) resembles the root of a land-based plant and plays the important role of anchoring it to the growth substrate (e.g., boulders for natural stocks and polymer substrates for cultivated seaweed [113]), though the contribution of the holdfast to nutrient uptake is insignificant. Above the holdfast is the cylindrical stipe, which imparts considerable flexibility to the blades to allow movement with water fluxes resulting from tides, waves, and local currents.

In high-energy waters, drag forces imposed on *S. latissima* may sweep algae away from their original habitat or even shred the algae, leading to mortality (linkage i in Figure 11). Also, water motion indirectly impacts the growth of *S. latissima* by affecting the nutrient uptake that takes place in the diffusion boundary layer close to the surface of *S. latissima* [66] (linkage j in Figure 11).

Water motion is one of the major drivers for the morphological diversification of *S. latissima* ecotypes and is believed to be a growth and survival strategy to modify the water flows over the algae [70] and thus reduce drag stress and diffusion stress [66]. In open water with great exposure, the blades tend to be more smooth and narrow to appropriately match forces, while, in low-energy environments, the blades appear to be wide, thin, and ruffled at the edges to increase the turbulence near the surface and improve nutrient exchange [114]. Interestingly, *S. latissima* specimens found in the Faroe Islands have hollow stipes, which seem to function similarly to the air bladders of *Fucus vesiculosus*, i.e., to increase the buoyancy of the algae [115].

##### Biofouling 

*S. latissima* is involved in symbiotic and parasitic associations that naturally occur in the aquatic ecosystems. Among the biotic stresses faced by *S. latissima*, including competition within populations for light and nutrients, grazing pressure, and so forth, biofouling receives the most attention due to its commonness and timing [116]. Epiphyte communities mainly consist of bryozoans but also include barnacles, blue mussel juveniles, and filamentous algae [55]. Particulate matter, including waste feed and feces released from fish culture, is also an important contributor to the “coating” of seaweed [117].

Biofouling takes effect through a variety of mechanisms and can lead to a significant decrease in growth, loss of biomass, and, in worst-case scenarios, defoliation of kelp beds [118,119]. In the case of grazers, the coverage of the epiphyte works as an enhancer of natural tissue loss. Biofouling also indirectly affects the growth of *S. latissima* by intensifying the stresses associated with other extrinsic factors; for instance, heavy fouling deprives sporophytes of irradiance [75] (linkage k in Figure 11). In addition, the settlement of epibionts, such as bryozoans, filamentous microorganisms, and larvae of barnacles, exerts mechanical forces on macroalgae that render them brittle [70]. This facilitates defoliation during mechanical events, such as storms and strong currents [57], and increases the risks of apical blade necrosis [34], therefore causing severe biomass loss. The impacts of epiphytes on the content and composition of compound groups in *S. latissima* have been investigated (e.g., amino acids [55] and phenolic compounds [64]). However, no clear correlations have been identified.

### 3.3. Applications of Extract Products and Associated Revenues

Algae biomass can be consumed/processed as a whole (e.g., as food/for soil conditioning and energy production; conversion pathways b and c-1 in Figure 13). It can also be fractionated into different constituents, which are then upgraded separately [45,120] (conversion pathways a, c-2, d, and e in Figure 13).

Alginate, mannitol, fucoidans, laminarin, proteins, and lipids are identified as the top value-added algal products obtained from *S. latissima*. Knowledge of the content of these components may improve the profitability of all actors in the macroalgal bioeconomy. The profitability depends on the biomolecular composition of the fresh biomass, which varies with time of harvest, and the market price of output products (Equation (Equation 1) and Figure 14). The following descriptions and the summary in Table 2 are not exhaustive but intended to highlight the most recent and potential applications of seaweed extracts as biorefinery products.

#### 3.3.1. Functionalities and Applications of Seaweed Biomolecules

Alginate is renowned for its gelling, emulsifying, thickening, and stabilizing properties [126] and has been long standing in the global hydrocolloid market, with a sales value of 345 million USD in 2015 [20]. Its conventional applications are primarily in the food industry, including suspension agents in salad dressings, thickeners for sauces, stabilizers for ice creams, creators of creamy long-lasting froth for beer [127], and binders for the restructuring of certain food, such as crab sticks and reconstructed meat [128,129,130]. With advances in technology, innovative applications have been developed on the basis of alginate’s biocompatibility and hydro-jelling properties. It is employed in pharmaceuticals and nutraceuticals (e.g., seamless alginate capsules [131], disintegrators [132], and wound dressing [133]), textile printing, welding, and so forth.

Fucoidans are favored for health-improving applications, such as nutraceuticals and dietary supplements [134], as they exhibit a number of pharmaceutically interesting biological activities, including anti-inflammatory [135], antitumor [136], antiviral [137] and anticoagulant [44] abilities. Concurrent with the increasing number of research publications documenting the positive health effects of fucoidans, there is a booming trend in the patent market as well. To name a few, fucoidans are used as an ingredient in pharmaceutical compositions [138], as a binder for surface coating [139], as a food and beverage additive [140], and as an exhaust-gas cleaning agent [141]. Currently, fucoidans extracts are available from the global marketplace (e.g., fucoidans extracts produced by Nutramara from Ireland (https://www.nutramara.com/fucoidan), by Sigma-Aldrich from the United States (https://www.sigmaaldrich.com/catalog/product/sigma/f5631?lang=en&region=US), and by Kanehide Bio from Japan (https://www.kanehide-bio.co.jp/product/en/product_fucoidan.html)) and ready for use in cosmetics, functional foods, and dietary supplements, as well as for inclusion in pet, livestock, and aquaculture feed supplements. However, it has not yet been developed as a regulated therapeutic either within biomaterials or via direct administration [142].

Similar to fucoidans, laminarin is increasingly recognized for its biofunctional activities and potential uses, such as a zinc oxide substituent in pig feed supplements [143], a therapeutic agent with immunostimulatory and anti-inflammatory properties [144], and an alternative fungicide to sulfur to be applied in vineyards [145]. Prospects for laminarin in the above-mentioned applications have also been realized by industries, as revealed by the inclusion of laminarin as an ingredient in many “a composition of matter” patents; for instance, in a mix for feed [146],as a composition useful for inhibiting skin-aging [147], and as a composition for food or healthcare products for regulating intestinal micro-ecology [148]. Despite the nutritional and therapeutic potential of laminarin, its development as a medical application is currently still in the clinical trial phase.

Mannitol has been extensively used in sweets and low-calorie foods [86]. Moreover, due to its favorable biological and chemical properties (non-hygroscopic, biocompatible, biodegradable, etc.), mannitol is a basic product widely used in chemical, pharmaceutical, and medical industries, such as in pharmaceutical formulations of chewable tablets and granulated powders [149], mannitol-based polyesters [150], and an osmotic diuretic for intoxication therapy [151]. Additionally, mannitol was found to be effective as an innovative carbon source in the partial nitrification process for wastewater treatment, facilitating nitrogen removal while remarkably reducing N${}_{2}$O emissions [152]. It is noteworthy that brown seaweed is not the only source of mannitol. Mannitol is abundant in essential agronomic and horticultural species [153]. Commercially available mannitol is commonly produced by industrial synthesis (i.e., hydrogenation of fructose), biosynthesis (i.e., fermentation by microorganisms, including bacteria, yeast, fungi, lichens, and algae), and natural extraction (e.g., from seaweed by Soxhlet extraction) [154].

Algal proteins and lipids receive less attention than carbohydrates in the field of single-component extraction due to their low content and a broad spectrum of existing alternative sources. However, along with the expanding global population, main nutrients, including proteins and lipids, will be in short supply in the future; in this context, algal proteins and lipids are strong candidates for fulfilling future generations’ nutrient demands.

Firstly, algal protein contains all the essential amino acids and a favorable amino acid profile [62], which makes algal protein comparable to that from soybean and other sources, such as egg, whole wheat, and rice. Therefore, algal protein can serve as a desirable alternative, especially for vegans, for whom eggs and dairy whey protein are not suitable. Beyond the provision of basic nutritional values, algal proteins can serve as potential substrates for the generation of bioactive peptides, which are particular amino acid sequences that become active when released from the parent protein by hydrolysis or fermentation [155]. Moreover, algal protein is a rich source of aspartic and glutamic acid, which are reported to be one of the important contributors to the umami taste [156]. The salt of glutamic acid, e.g., monosodium glutamate (MSG), is commonly applied in food processing as a flavor enhancer, and it is classified as a food additive by the European Union (code: E621) [157].

Despite having relatively low tissue content, *S. latissima* has a favorable fatty acid profile with high proportions of long-chain polyunsaturated fatty acids (LC-PUFA), especially stearidonic acid (SDA; C18:4*n*-3), arachidonic acid (ARA; C20:4*n*-6), eicosapentaenoic acid (EPA; C20:5*n*-3), and docosahexaenoic acid (DHA; C20:6*n*-3) [158]. This makes *S. latissima* an alternative source of LC-PUFA other than traditional vegetables [159]. Moreover, algal lipids have the right balance of *n*-3 and *n*-6 PUFA [47,69]. Beneficial *n*-3 fatty acids are known to contribute to the prevention of cardiovascular diseases [160,161], and EPA and DHA are particularly effective in the prevention and treatment of depression [162]. In contrast, *n*-6 fatty acids tend to promote tumor growth and inflammation [163]. Therefore, lipids recovered from seaweed with a suitable ratio between *n*-3 and *n*-6 PUFA are fit for use as nutraceuticals in the functional food industry [164] or in the pharmaceutical industry for various uses, such as a cell activator [165] for treating allergic disease, cardiac disease, diabetes, and cancer.

#### 3.3.2. Revenues Associated with Biomolecules Extracted from *S. latissima*

Figure 14 presents the projected revenues of the extracts of polysaccharides, lipids, and proteins that are produced from seaweed biomass, which is harvested at different times. The box-and-whisker plots in Figure 14 summarize the spatial and seasonal variations by showing minimum, first quartile, median, third quartile, and maximum content of biomolecules in *S. latissima* and associated revenues. The central box represents the likely range of variation (from the first quartile to the third quartile), and the band inside the box shows the typical value (i.e., the median). The whiskers above and below the box display the full range of variation (from min to max).

As displayed in Figure 14, the associated revenues can vary greatly with harvest time (shown by the bands inside the boxplots along *x* axis). Revenues from alginate extracts are constant over the entire period of all potential harvest months. The only exception is August where the median is 20% lower than the revenues seen in other months. Revenue from mannitol extract is fairly low and stable in comparison with revenues from other sugar extracts, and the best harvest time is July. Fucoidan extracts have the highest revenue during the period from April to June among all sugar extracts, and September seems to be the best time for fucoidan extracts according to the median value; however, the wide range shown by the whiskers indicates potential uncertainty. Laminarin extract appears to be economically promising from May onward, and July is the most suitable month according to the relatively high median and the moderate uncertainty. Revenues from protein extracts slightly fluctuate throughout the whole period, and September appears to be the most suitable harvest time if proteins are the target output product.

However, the wide variation caused by locality (shown by the boxes and whiskers) makes it impossible to draw generalized recommendations for optimal harvest times for one specific output product at different production sites. While seasonality does exert an influence on the trend, local environmental parameters appear to play a crucial role in determining the degree of the effect of seasonality on the biochemical composition and associated revenues.

## 4. Discussion

### 4.1. Farming Practices for Biomass Value Optimization

Over time and multiple generations of adaption, the impacts of large-scale regulating factors (e.g., light, temperature, and salinity regimes) on *S. latissima* have become trivial owing to light-related [92], temperature-related [166,167], and salinity-related ecotypic differentiation [102]. One typical example is illustrated by *S. latissima* cultivation in Spain and Norway; both countries produce similar yields, although the oceanographic conditions (especially temperature) in Spain allow a growing period of only 5 or 6 months, whereas the growth phase in Norway can last from August to the following year’s autumn [4].

Despite adaptions, the results presented in Section 3.1 reveal a tendency for different growth patterns between *S. latissima* specimens with different life histories. This indicates the essential role of life history, which basically describes the interactions between endogenous rhythms and local-scale extrinsic factors, in determining the biomolecular composition of *S. latissima*. On the basis of observed seasonal variation in the biomolecular composition (Section 3.1) and the current technology readiness level (TRL) of seaweed biorefinery, we foresee that a phase I biorefinery, in which the seaweed biomass is utilized for producing one major product and several co-products (conversion pathways a and b in Figure 13), is currently still the most economically feasible approach for businesses. However, local seaweed production and side-stream valorization are obtaining increased interest from local governments, businesses, and not least consumers (e.g., conversion pathways c, d, and e in Figure 13). For this reason, cascade utilization of locally produced seaweed represents a potential business opportunity expected to replace current phase I biorefinery in the future [168,169,170].

#### 4.1.1. Encourage Growth by Careful Timing of Production Activities

Deployment—the starting step of offshore farming—is suggested to take place as early as is allowed by local conditions, e.g., early autumn in Norwegian waters [119] and early winter in Spanish waters [4]. Early deployment provides *S. latissima* with sufficient time to build its internal N reserve, and this internal reserve enhances the growth rates of *S. latissima* in the summer months when sporophyte growth is potentially restricted by a N-deficient aquatic environment [36,108,171]. Late deployment and the resultant lack of endogenous N reserve build-up are likely to reduce the growth and development of *S. latissima*, as implied by the delayed peak and reduced maximum levels of proteins and alginate, respectively (seasonal and spatial variations presented in Section 3.1.2), which are fundamental building blocks for cellular complexes and cell wall construction.

At a later stage of farming, the timing and frequency of harvest are key factors in boosting the overall productivity of seaweed biomass, taking into account the downstream biorefinery target output products. Multiple harvests in a single growing season [3] and continuous production across cultivation years [172,173] are two practices based on the natural growth patterns of *S. latissima*: i.e., as a perennial, *S. latissima* sheds its apical blade, which regrows in the next growing season [174]. The hypothesis that partially cutting mature apical blades does not harm the algae but stimulate its (re)growth was verified by the increased biomass yield obtained in the second cultivation year of *S. latissima* in Hjarnø, Denmark [172]. Moreover, multiple harvests from the same batch deployment represent a potential increase in resource efficiency: i.e., cultivators can increase the biomass yield without additional inputs or investments on seed line production, which covers spore isolation, development, settlement, and germination, as presented in Figure 12.

#### 4.1.2. Improve Photosynthesis by Smart System Design

Photosynthesis is the most important life-sustaining process for *S. latissima* as an autotroph. Increased irradiance availability and photon utilization is a potential direction for intensifying *S. latissima* mariculture. This can be realized by maintaining sufficient space between individual sporophytes on seeded lines [4] and adding water-mixing regimes near cultivation systems [175]. The former avoids self-shading, and the latter resolves natural constraints caused by water turbidity.

Besides phytogeography and productivity, sunlight is the primary factor that determines the composition of *S. latissima*. Mannitol, which is a primary product of the photosynthesis process, is a central compound in carbon metabolism, as well as in the transportation and distribution of the organic assimilates in *S. latissima*. Mannitol metabolism regulates the balance between mannitol and F6P and furthers other related pathways of, for example, alginate and fucoidans. Therefore, smart system designs with a focus on improving photosynthesis not only contribute to the overall growth of *S. latissima* but also establish the basis (i.e., production of mannitol) underpinning the formation of other biomolecules, e.g., proteins, in *S. latissima*.

#### 4.1.3. Achieve the Desired Biomolecular Composition by Flexible At-Sea Management

Following from the above discussion is the understanding that mannitol is involved in different biochemical formation pathways for various biomolecules in *S. latissima*. These pathways are interchangeable processes depending on local environmental conditions. As such, relative tissue content of biomolecules show different trends, and their maxima and minima seldom coincide (presented in Section 3.1).

To obtain the desired biomolecular composition profile, regular sampling of biomass and concomitant monitoring of local environmental conditions (hydrographic and weather data in particular) are necessary. Seaweed farm operators can take immediate actions on the basis of collected real-time information on the growth performance of *S. latissima* and extrinsic factors in the field, which influence mannitol allocation and the formation of the biomolecules of interest.

Flexible systems, such as a floating rig, whose position can be adjusted horizontally and vertically [176], can be of great use under adverse conditions, such as hyposalinity, nutrient limitations, or high-energy water movement during stormy periods. When confronted with decreased salinity in the upper water column caused by the inflow of freshwater or lowered vertical mixing, lowering the cultivation system to deeper water columns, where salinity levels are generally higher, could be a solution. In the summer periods, during which ambient nutrient levels are commonly low and upwelling takes place simultaneously, a cultivation system moved to an upwelling zone may benefit from the nutrient-rich waters flushed to the surface by waves. However, water motions are not always beneficial to the growth of *S. latissima*. In stormy events, in addition to the great hydrodynamic forces, the circular movements of waves in the open sea expose fronds of *S. latissima* to more than one wave at a time and increase the destructiveness of the wave motions. Given the tendency of the circular movements’ radius to decrease with depth, moving downward may serve as a solution to survive strong wave forces and avoid large-scale seaweed loss.

Moving a seaweed cultivation system closer to available aquaculture in the vicinity can also serve as an approach to providing cultivated seaweed with favorable conditions or alleviating environmental stresses. For example, in mussel farming, moving the seaweed cultivation rigs to where mussel lines are situated could help improve the accessibility of seaweed to the necessary irradiance, which is limited by algal blooms or poor water mixing. This is because mussels are efficient suspension feeders and promote water transparency and light penetration [177].

### 4.2. An Integrated Framework to Maximize Ecosystem Services Delivered by Seaweed Cultivation and Harvest

Owing to its restorative and regenerative properties, green-engineered seaweed production can be regarded as a nature-based solution to environmental degradation and resource over-exploitation, as well as a resource center (i.e., transforming emissions into complex biomolecules for manufacturing a broad range of biobased products) to reconcile the needs of a growing population with the resulting impact on our environment.

Currently, only a few offshore seaweed farms have been implemented at a business scale, and they are primarily located in Europe; to name a few, Ocean Rainforest from the Faroe Islands (http://oceanrainforest.com/), Seaweed Energy Solutions from Norway (http://seaweedenergysolutions.com/en), Seaweed Harvest Holland from the Netherlands (https://seaweedharvestholland.nl/archief.html), and Islander Rathlin Kelp from Ireland (https://islanderkelp.com/process/). Further development of the macroalgal economy requires policy instruments to support the growth of an environmentally resilient blue economy and reduce the vulnerability of the seaweed production business to changing market conditions and climate-driven regime shifts. Local authorities have the opportunity to promote seaweed farming as an ecosystem health-improving instrument and align the development of macroalgal businesses with long-term sustainability.

We therefore propose an ecosystem-based management framework (Figure 15) consisting of a triple-helix partnership between industry (seaweed cultivation and biorefinery companies), local authorities, and research institutions. This partnership acts as an innovation hub [178] and involves all stakeholders in the implementation of an adaptive water quality monitoring program with the common objective of achieving a balance between marine ecological integrity and ecosystem service utilization.

#### 4.2.1. Macroalgal Business Actors: Opportunities and Challenges in Developing the Blue Economy

Seaweed production provides “blue biomass” that contains valuable biomolecules, which are converted from assimilated excess aquatic nutrients and atmospheric CO${}_{2}$ through photosynthesis and metabolism. If managed appropriately, seaweed, as a renewable resource, is in no danger of being overexploited [17]. On the basis of observed trends in the changing biomolecular composition profile (Section 3.1) and the current TRL of seaweed biorefinery, we suggest that a phase I biorefinery (i.e., a single raw material is utilized for producing one major product and several co-products) is the most applicable for business in practice in pursuit of high revenues from seaweed extracts.

The prices of extract products derived from seaweed biomass are remarkably high (presented in Table 2) in comparison with products generated in other utilization pathways, such as utilizing entire biomass for food and feed or converting it to biofuels. With extra effort placed on product refinement, the prices of extract products can further rise. However, it is noteworthy that despite the splendid future in which all actors along this value chain may benefit from the high revenue (projections are shown in Figure 14), there arises the need for more sophisticated processing and stricter quality control.

Economic feasibility of the supply chain from a seaweed nursery, to deployment, to growth, to harvest, and to product extraction is obtained under the following circumstances: First, for the seaweed farmer, the revenue of the harvested seaweed is exceeded the cost of biomass production; and second, the costs of the feedstock and the down-stream technologies used to extract high-value biomolecules are exceeded by the revenue from the sales of final output extract products [179]. A knowledge-based market pricing of the seaweed as feedstock for biorefineries, i.e., according the biomolecular composition at the to time of harvest, may support optimal business design of the upstream cultivators according to seasonal market demand. For the downstream part of the value chains, i.e., the biorefinery business, knowledge of the cost of revenue may help set a maximum feedstock price. Lastly, quantification of the environmental sustainability, e.g., the net CO_2_ footprint, of the whole value chain needs to be ensured by policy instruments. As such, a triple-helix partnership concept (Figure 15) to support long-term local sustainable business relationships along the whole value chain, driven by single and multiple business actors in a collaborative process is needed [180].

In addition, the seaweed production business is subjected to the challenges of changing field conditions in the global context of climate change. Unusual weather patterns have been witnessed worldwide, confirming the fact that our global environment has stepped into a transition phase, and these changes will inevitably have direct and indirect impacts on the marine ecosystem [181]. Including global warming-related phenomena, such as the prolonged period of ice cover, periodic high temperatures during the summer, increased freshwater flows from glaciers/melting snow, and heavier precipitation [182,183], as well as the changing ultraviolet (UV) radiation regime, which is associated with stratospheric ozone depletion [184], these climatic changes may trigger modifications to biological, chemical, and physical processes at different aquatic trophic levels. In turn, these changes can reduce marine ecosystem resilience and thus pose threats to the production of seaweed with replicable quantity and quality. Though macroalgae are capable of acclimating to environmental stresses to some degree, their overall tolerance level is presumably reduced when confronting multiple stresses simultaneously, and they can become locally extinct if the stresses are persistent and keep developing.

Close collaboration with academia and local authorities is therefore of significance for actors in the seaweed business to improve financial viability, with the former providing research-based decision support on the best available farming practices and biorefinery technologies to optimize productivity and the latter applying regulatory instruments to provide financial aid and drive eco-efficient investment behavior.

#### 4.2.2. Local Authorities: Valorization of Seaweed Farming’s Effort in Water Quality Restoration

Massive changes to biogeochemical flows due to anthropogenic activities have led to an imbalanced distribution of N and P among multi-spheres; namely, a biosphere lacking in nutrients and a hydrosphere turning pools of excess nutrients. Nutrient flows entering the marine ecosystem come from numberless sources, such as runoff from agricultural fields and sewage effluents, apart from coastal fish or mussel farms [185]. Also, there exists an internal loading of historical phosphorus emissions that have been retained in the sediment since the start of industrial times and that are sometimes released to the hydrosphere due to changes in water motions or ocean chemistry.

As a green-engineered system, which mimics nature’s supporting service of nutrient cycling, seaweed production is a promising instrument for water quality restoration [10]. Even though results of the most recent studies evaluating the potential of *S. latissima* farming as a short-term nutrient-scavenging tool at the local level have generally shown that it is impractical to capture fully equivalent soluble inorganic nutrient emissions from the established fish farming sector or IMTA system due to different growth patterns of fish and *S. latissima* [119,186], as well as the wide distribution of nitrogen emanating from fish culture [117], partial nutrient sequestration of seaweed production can be advantageous [187]. Moreover, global imbalance of the biogeochemical nitrogen and phosphorous cycles can only be mitigated through an intensified implementation of seaweed cultivation, e.g., increasing the productivity of local production or cultivating seaweed at wider geographical scales [117], if seaweed cultivation is to be used as an instrument for long-term circular nutrient management [14].

Seaweed farming has been promoted within existing aquaculture sectors of several European countries as a starting point in the move toward large-scale eutrophication mitigation, which is one of the goals set by directives in the European Union (EU) (e.g., the Water Framework Directive (WFD)) and at the regional level (e.g., the Baltic Sea Action Plan (BASP)). As an example, through the establishment of N-credits for seaweed farming, Denmark’s aquaculture business actors are motivated to incorporate seaweed farming in their conventional business scheme to compensate for the N-quota that limits their fishery activities [188]. In northwestern Spain, seaweed cultures are encouraged to grow on used mussel farming rafts [4].

To promote the future expansion of seaweed farming, which is currently in its infancy, we suggest that local authorities provide financial support and economic incentives, e.g., through a subsidy scheme that is based on the yearly kinetics of seaweed mitigation efforts in water catchment areas.

#### 4.2.3. Local Authorities: Alignment of the Macroalgal Bioeconomy with Long-Term Sustainability and Resilience

Ecosystem services delivered by macroalgal culture are not limited to nutrient cycling, but also include fixing atmospheric carbon into biomass, providing habitats for low-trophic-level sea creatures, and contributing to coastal secondary production [189], among other functions. Through bottom-up trophic cascades (from local to global scales), seaweed culture has a far-reaching influence on marine ecosystem dynamics and planetary resilience.

If the cultivated seaweed is harvested and utilized, the effect of carbon sequestration may take place beyond the farm gate through the substitution of commercial products that possess similar functions but have poorer environmental performance. For instance, macroalgal food products can complement traditional agriculture—which has already undergone intensification and possesses limited potential for expansion—to meet the increasing nutritional needs of the growing global population and provide humanity with the health-promoting functions afforded by the bioactive molecules in algal biomass. As a further example, partially replacing conventional feed for ruminant animals with seaweed can achieve reductions in enteric methane emissions [190,191,192]. Besides the ecological benefits, innovative algal products open up new commercial fields with economic activities that bring social benefits, such as job creation.

While these exciting signs of progress reinforce the belief in a promising macroalgal bioeconomy, it is important to consider the kinetics of the degradation of macroalgal products. Taking carbon fixation as an example, when we appraise the bioassimilation of atmospheric carbon by seaweed production, it is necessary to pose a question: how long can the captured carbon stay in the biosphere, lithosphere, or other compartments before returning to the atmosphere? If considering only the CO${}_{2}$ footprints directly linked to macroalgal value chains (i.e., excluding the substitution effects of algal products), converting seaweed biomass to bioethanol and foodstuffs are not likely to achieve long-term CO${}_{2}$ sequestration due to the fast degradation of these products and the subsequent quick release of carbon back into the atmosphere [15,193]. The application of seaweed-derived biochar to soil is a promising candidate for fixing atmospheric CO${}_{2}$ in the lithosphere in the long term and in a significant amount, i.e., 50% of initial C can be retained in soil in comparison with burning (3%) and biological decomposition (from <10%–20% after 5–10 years) [194]. Additionally, biochar enhances soil fertility by returning P [195]. Moreover, there can be other types of environmental burdens introduced at processing, transportation, and other life cycle stages of algal products [123].

The stepwise development of the cumulative CO${}_{2}$ footprint and other environmental impacts along macroalgal value chains shows a need for regulatory instruments to align the development of the macroalgal bioeconomy with long-term sustainability and resilience, which are stressed by sustainable development goals (SDGs), the EU’s Blue Growth strategy, and other political strategies.

#### 4.2.4. Research Institutes: Knowledge Sharing between all Stakeholders according to a Monitoring Program

Ecological and physiological academic knowledge of *S. latissima* and its interactions with natural and anthropogenic factors is needed for business actors to optimize biomass utilization and value creation and for local authorities to construct a more profound appraisal and performance management framework for developing the macroalgal bioeconomy. Besides scientific research efforts, knowledge inputs from business actors and local authorities are needed. The aggregation of all stakeholders’ inputs is expected to create synergy effects and can be achieved by establishing a science-based water quality restoration monitoring program.

The role of seaweed farmers in the monitoring program is to document the field conditions and growth performance of *S. latissima*. To do so, during the early stage of site selection, cultivators need to consider the availability of monitoring stations in the vicinity of planned deployment sites and the accessibility of marine weather and water forecast data. Reporting real-time growth performance and local conditions can facilitate scientific research on the interaction between *S. latissima* and abiotic and biotic environmental factors throughout its life history. Moreover, the documentation provides evidence of the contributions of seaweed farming to improving water quality, which helps local authorities fulfill obligations under the WFD and BASP.

Regarding documentation, we propose firstly a comprehensive common database covering the local environmental conditions in the field, the biomolecular composition of *S. latissima*, and important characteristics revealing the functional properties of biomolecules, e.g., the M/G ratio in Alginate [196], DS of fucoidans [13], digestibility of proteins, and bioavailability of amino acids [62]. Secondly, we advise harmonized methods for quantifying yields of different biomolecules. Unifying these methods is necessary for making comparisons across sites and between aqua-based and land-based crops, e.g., areal yield expressed in kg biomolecules per ha cultivation area [46].

As a result of the shared knowledge and funding support from seaweed cultivators, scientific workers can predict the biological responses of cultivated seaweed to the up-coming changes in extrinsic factors on the molecular level, on the basis of which seaweed cultivators and biorefinery operators can take immediate actions, such as changing the position of the cultivation system, conducting an early harvest, and changing combinations of biorefinery conversion pathways. Furthermore, seaweed business actors are provided with important information, including mapping of species of economic interest [130], guidance for seaweed production [90,197], and forecasts of the macroalgal bioeconomy [198].

## 5. Conclusions

A solid understanding of seasonal compositional variation according to local site characteristics can improve the net economic viability of seaweed bioeconomic businesses through the optimized productivity of known composition. Seasonal and locational optimal strategic seaweed cultivation and biorefinery business plans may only be obtained through knowledge of the inter-correlations between extrinsic and endogenous metabolic factors. Therefore, we propose an ecosystem-based framework for an adaptive water quality restoration program that is based on a triple-helix partnership between local authorities, research institutions, and the seaweed cultivation and biorefinery companies for sharing knowledge to achieve political alignments, improve ecological and physiological knowledge, and develop macroalgal bioeconomy, respectively, and for pursuing the common goal of reconciling the marine ecosystem without negatively impacting human use.

## Figures and Tables

**Figure 1 marinedrugs-17-00107-f001:**
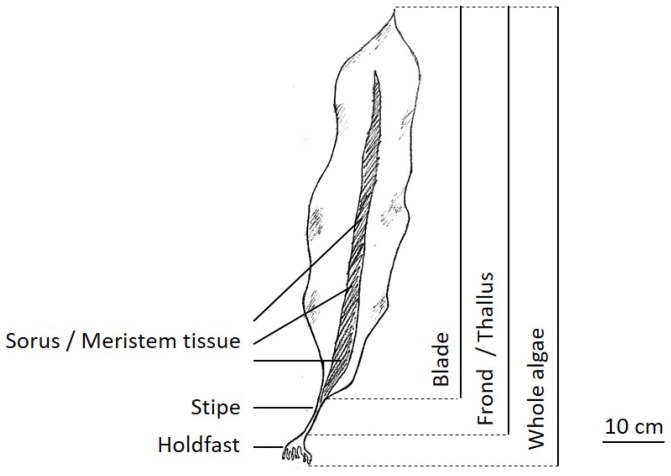
Sketch of *S. latissima*.

**Figure 2 marinedrugs-17-00107-f002:**
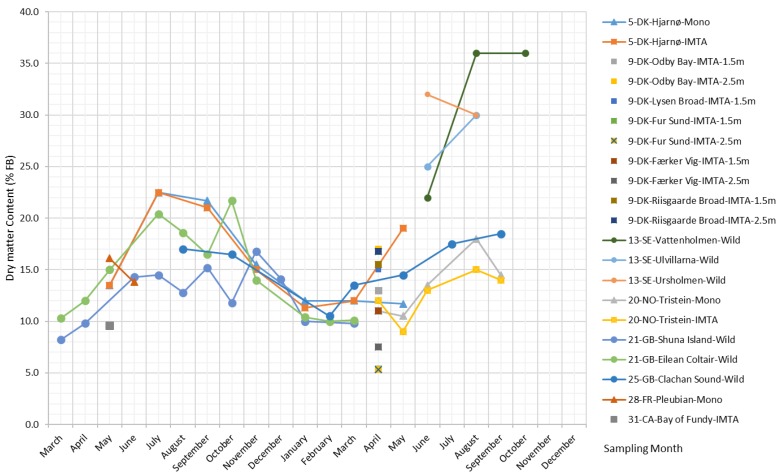
Dry biomass content expressed as the percentage of fresh *S. latissima* biomass (% fresh biomass). The legend is displayed as [Reference No.]-[Country Code [33]]-[Sampling site]-“Cultivated” ([Mono] or [IMTA]) or “Wild” ([Wild])-[Cultivation depth]. Reference Nos. match those in the table in the Appendix A. Data labels: circles (wild stock), triangles (monoculture), and squares (IMTA) represent the origins of the sampled biomass.

**Figure 3 marinedrugs-17-00107-f003:**
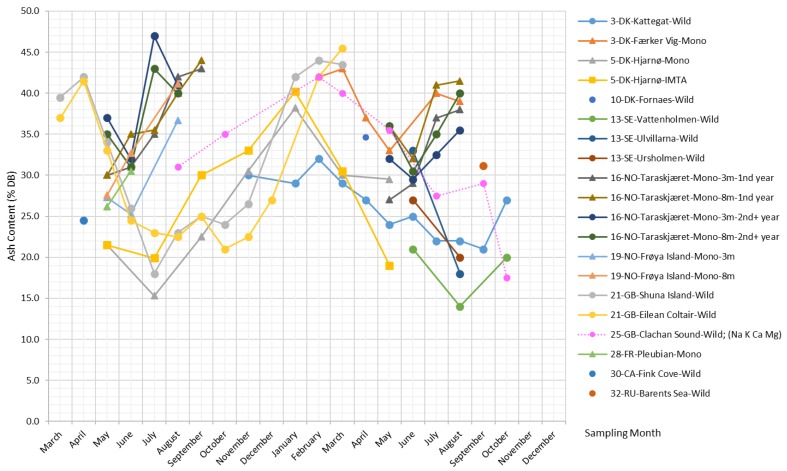
Ash content expressed as the percentage of dry *S. latissima* biomass (% dry biomass). The legend is displayed as [Reference No.]-[Country Code [33]]-[Sampling site]-“Cultivated” ([Mono] or [IMTA]) or “Wild” ([Wild])-[Cultivation depth]-[Maturity]. Reference Nos. match those in the table in the Appendix A. Data labels: circles (wild stock), triangles (monoculture), and squares (IMTA) represent the origins of the sampled biomass.

**Figure 4 marinedrugs-17-00107-f004:**
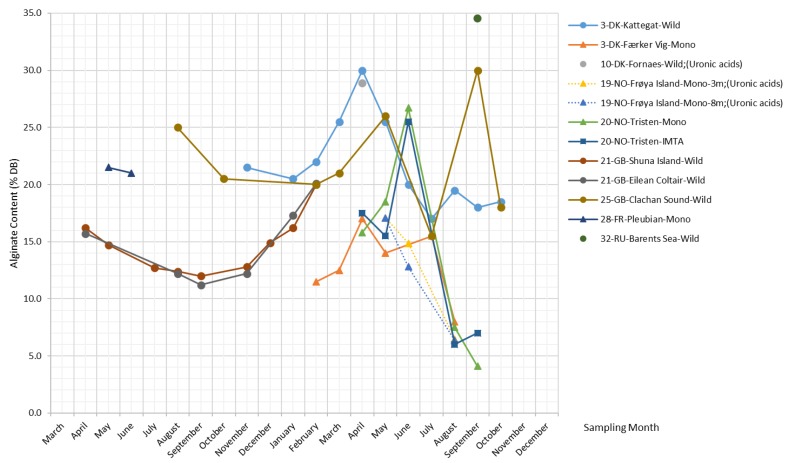
Alginate content expressed as the percentage of dry *S. latissima* biomass (% dry biomass). The legend is displayed as [Reference No.]-[Country Code [33]]-[Sampling site]-“Cultivated” ([Mono] or [IMTA]) or “Wild” ([Wild])-[Cultivation depth]. Reference Nos. match those in the table in the Appendix A. Data labels: circles (wild stock), triangles (monoculture), and squares (IMTA) represent the origins of the sampled biomass.

**Figure 5 marinedrugs-17-00107-f005:**
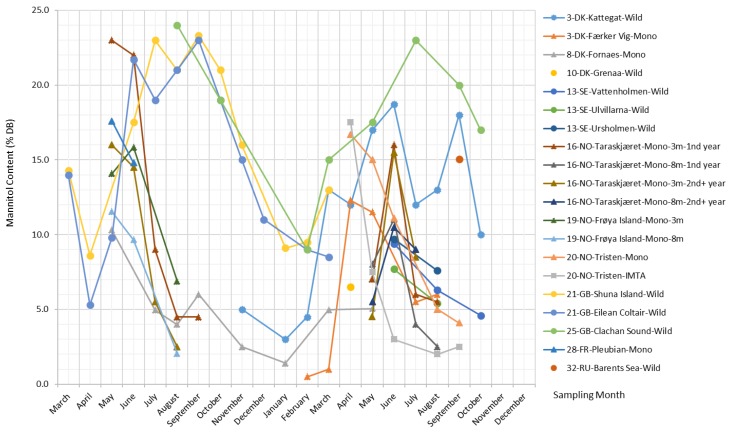
Mannitol content expressed as the percentage of dry *S. latissima* biomass (% dry biomass). The legend is displayed as [Reference No.]-[Country Code [33]]-[Sampling site]-“Cultivated” ([Mono] or [IMTA]) or “Wild” ([Wild])-[Cultivation depth]-[Maturity]. Reference Nos. match those in the table in the Appendix A. Data labels: circles (wild stock), triangles (monoculture), and squares (IMTA) represent the origins of the sampled biomass.

**Figure 6 marinedrugs-17-00107-f006:**
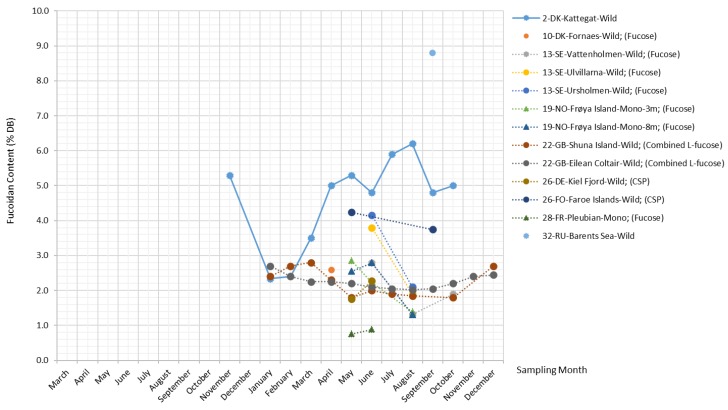
Fucoidan content expressed as the percentage of dry *S. latissima* biomass (% dry biomass). The legend is displayed as [Reference No.]-[Country Code [33]]-[Sampling site]-“Cultivated” ([Mono] or [IMTA]) or “Wild” ([Wild])-[Cultivation depth]. Reference Nos. match those in the table in the Appendix A. Data labels: circles (wild stock), triangles (monoculture), and squares (IMTA) represent the origins of the sampled biomass.

**Figure 7 marinedrugs-17-00107-f007:**
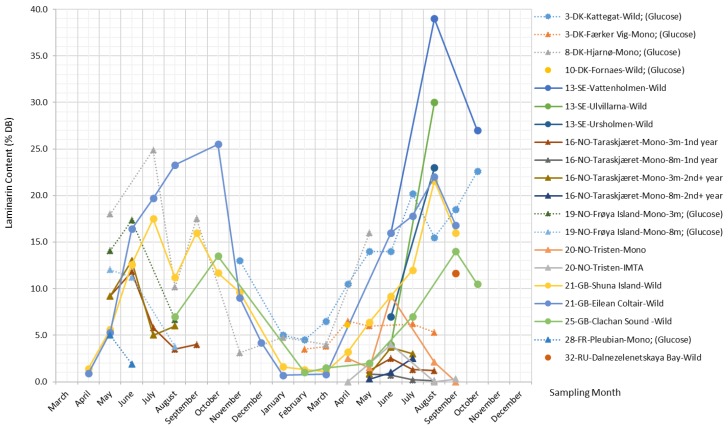
Laminarin content expressed as the percentage of dry *S. latissima* biomass (% dry biomass). The legend is displayed as [Reference No.]-[Country Code [33]]-[Sampling site]-“Cultivated” ([Mono] or [IMTA]) or “Wild” ([Wild])-[Cultivation depth]. Reference Nos. match those in the table in the Appendix A. Data labels: circles (wild stock), triangles (monoculture), and squares (IMTA) represent the origins of the sampled biomass.

**Figure 8 marinedrugs-17-00107-f008:**
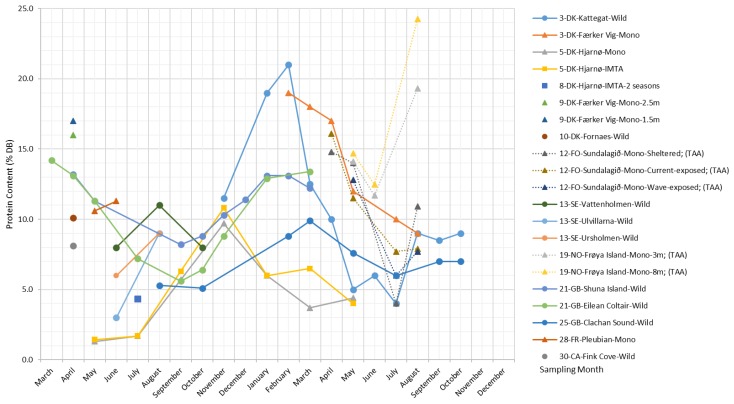
Protein content expressed as the percentage of dry *S. latissima* biomass (% dry biomass). The legend is displayed as [Reference No.]-[Country Code [33]]-[Sampling site]-“Cultivated” ([Mono] or [IMTA]) or “Wild” ([Wild])-[Cultivation depth]-[Maturity]. Reference Nos. match those in the table in the Appendix A. Data labels: circles (wild stock), triangles (monoculture), and squares (IMTA) represent the origins of the sampled biomass.

**Figure 9 marinedrugs-17-00107-f009:**
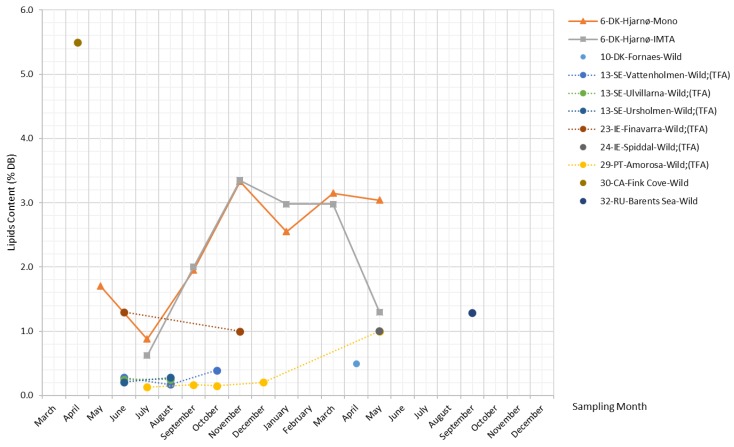
Lipids content expressed as the percentage of dry *S. latissima* biomass (% dry biomass). The legend is displayed as [Reference No.]-[Country Code [33]]-[Sampling site]-“Cultivated” ([Mono] or [IMTA]) or “Wild” ([Wild])-[Variants(Maturity, depth or exposure)]. Reference Nos. match those in the table in the Appendix A. Data labels: circles (wild stock), triangles (monoculture), and squares (IMTA) represent the origins of the sampled biomass.

**Figure 10 marinedrugs-17-00107-f010:**
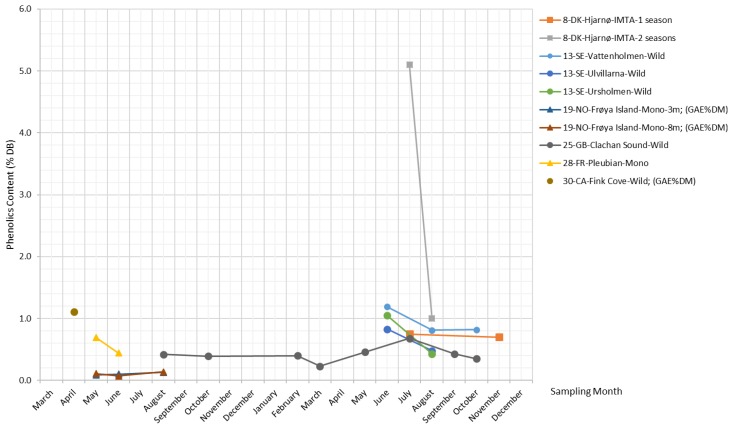
Phenolics content expressed as the percentage of dry *S. latissima* biomass (% dry biomass). The legend is displayed as [Reference No.]-[Country Code [33]]-[Sampling site]-“Cultivated” ([Mono] or [IMTA]) or “Wild” ([Wild])-[Cultivation depth]. Reference Nos. match those in the table in the Appendix A. Data labels: circles (wild stock), triangles (monoculture), and squares (IMTA) represent the origins of the sampled biomass.

**Figure 11 marinedrugs-17-00107-f011:**
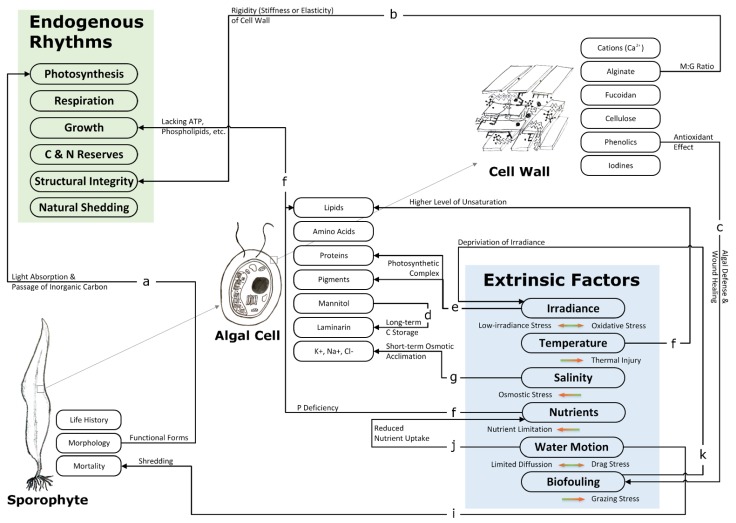
Potential interactions between key biomolecules of *S. latissima*, bioactivities, and extrinsic factors. The sketch of the cell wall structure is drawn on the basis of previous studies [43,49,66,67] with modifications. The lists of biomolecules, extrinsic factors, and bioactivities and the linkages between them are not exhaustive, but are illustrative of the potential correlations underlying the changing biomolecular profile observed for *S. latissima* samples (discussed in Section 3.2). The degree of pressure imposed by individual extrinsic factor on *S. latissima* is indicated by the color of the arrow, i.e., green (low) and red (high). The optimal values/ranges of each factor for the growth and development of *S. latissima* are summarized in Table 4.

**Figure 12 marinedrugs-17-00107-f012:**
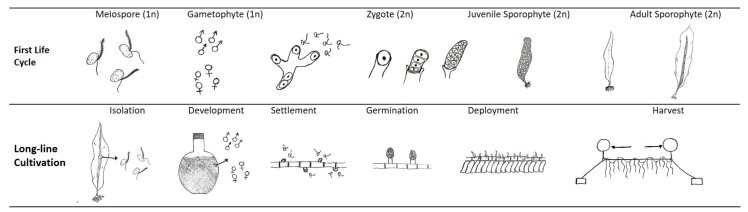
Life cycle of *S. latissima* and corresponding processes of the long-line cultivation method.

**Figure 13 marinedrugs-17-00107-f013:**
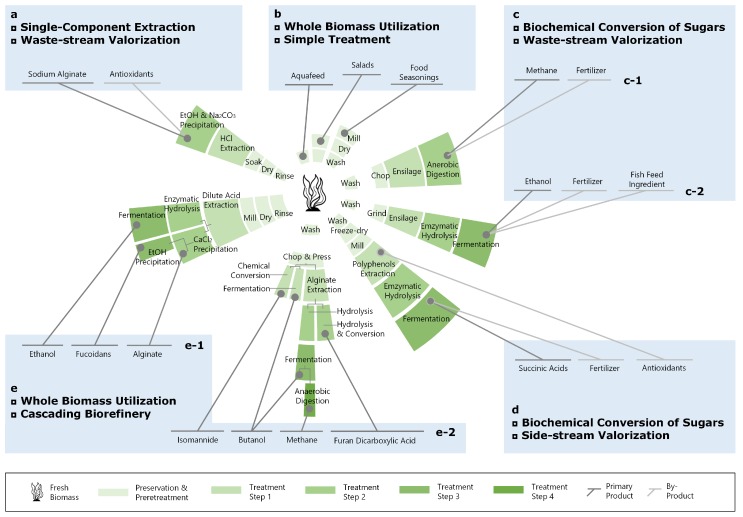
Examples of existing and innovative conversion pathways for seaweed biomass. Reference: a [121]; b [122]; c-1 [123]; c-2 [10]; d [7]; e-1 [124]; e-2 [125].

**Figure 14 marinedrugs-17-00107-f014:**
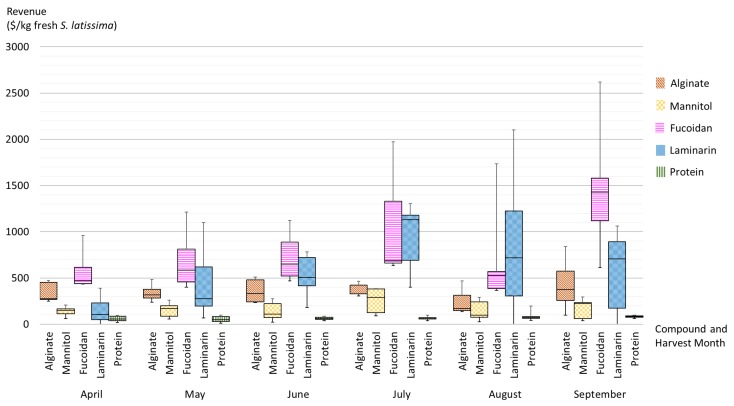
Boxplot of the seasonal variation in potential revenues associated with the seasonal variation in algal products extracted from 1 kg fresh *S. latissima* biomass harvested at different potential harvest time points (detailed calculation provided in the Appendix A).

**Figure 15 marinedrugs-17-00107-f015:**
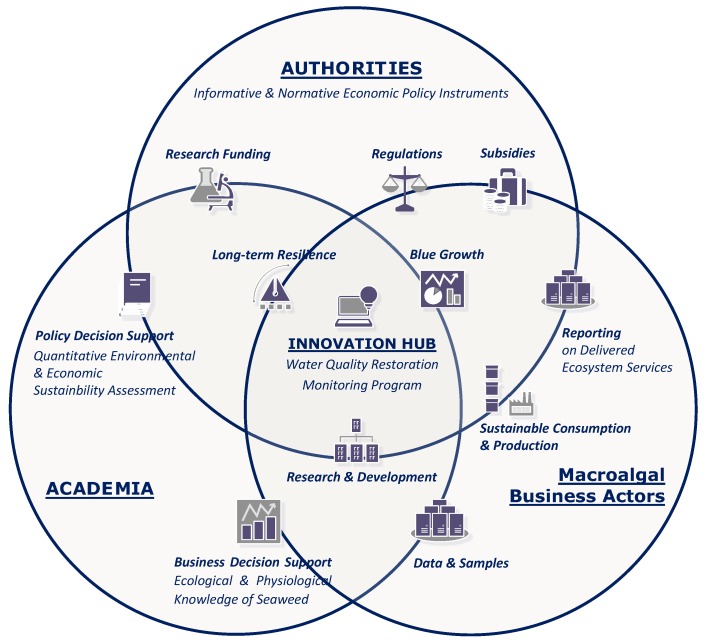
An ecosystem-based framework consisting of a triadic interrelationship between academia, industry and government with the common objective of implementing a sustainable blue economy delivering provisional (biorefinery products), supporting (nutrient recycling), and regulating (climate change mitigation) services, thereby restoring the balance between marine ecological integrity and ecosystem service utilization.

**Table 1 marinedrugs-17-00107-t001:** General information about *Saccharina latissima*.

Aspect	Description
Kingdom	Protista
Class	Phaeophyceae
Order	Laminariales
Genus/Family	Laminariaceae
Global distribution 1	North Atlantic Ocean and North Pacific Ocean
Habitat	Subtidal zone
Life cycle	2–5 years
Perennial	Yes

**Table 2 marinedrugs-17-00107-t002:** Market price of top value-added algal products that can be derived from *S. latissima* and their existing and innovative applications. Reference: a [18]; b [19]; c [20]; d [21]; e [22]; f [23]; g [24]; h [25]; i [26]; j [27]; k [28]; l [29]; m [30]; n [31]; o [32].

Products	Crude Product Price ($/kg)	Refined Product Price ($/kg)	Applications	Industry
Alginate	$6\;{}^{a}$; $12\;{}^{b}$; $14\;{}^{c}$; $24\;{}^{d}$	$162.8\;{}^{k}$; $290\;{}^{l}$	Emulsifier; Binding agent; Stabilizer; Coating materials; Wound healing	Food and beverages; Pharmaceutical and para-pharmaceutical; Processing
Mannitol	$6\;{}^{a}$; $7.3\;{}^{e}$; $11\;{}^{f}$	56 (≥98%) ^m^	Sweetener; Flavoring agent; Stabilizer; Diuretics	Food; Pharmaceutical; Medical; Chemical
Fucoidan	$12\;{}^{a}$; $150\;{}^{g}$; 175(30% or $40\%)\;{}^{h}$; $350\;(60\%)\;{}^{h}$;	$398,000\;(95\%)\;{}^{n}$	Functional food; Dietary supplements; Drug delivery; Cosmetics	Food and Beverages; Human health; Therapeutics
Laminarin	$14.5\;{}^{a}$; $51.72\;{}^{i}$; $1,738\;{}^{j}$	$98,200\;{}^{o}$	Feed supplements; Pesticide	Nutraceutical; Cosmetics; Agriculture and aquaculture
Protein	$5\;{}^{a}$		Feed additives; Functional food ingredients; Flavor enhancer	Agriculture and aquaculture; Processing food; Human nutrition
Lipids	$2\;{}^{a}$		Functional food	Nutraceutical; Pharmaceutical

**Table 3 marinedrugs-17-00107-t003:** Major carbohydrates in *S. latissima* and their functions, fundamental subunits, and formula.

Polysaccharides	Primary Function	Monosaccharides	Formula
Laminarin	Carbon storage	Glucose	C_6_H_12_O_6_
		Mannitol	C_6_H_14_O_6_
Mannitol *	Carbon storage; Osmoprotectant; Antioxidant	Mannitol	C_6_H_14_O_6_
Alginic acid	Cell wall structure	Mannuronic acid	C_6_H_10_O_7_
		Guluronic acid	C_6_H_10_O_7_
Fucoidan	Cell wall structure;	Fucose	C_6_H_12_O_5_
		Galactose	C_6_H_12_O_6_
		Mannose	C_6_H_12_O_6_
		Xylose	C_5_H_10_O_5_
		Arabinose	C_5_H_10_O_5_
		Glucuronic acid	C_6_H_10_O_7_
Cellulose	Cell wall structure	Glucose	C_6_H_12_O_6_

**Table 4 marinedrugs-17-00107-t004:** Overview of environmental factors that have profound effects on *S. latissima* and may cause stresses for *S. latissima* when they are far from their optimal values or ranges.

Factors	Stresses	Optimum	Value	Unit	Reference
Irradiance	Oxidative stress	Photosynthetic saturation level	around 150	μEm${}^{-2}$ s${}^{-1}$	[84]
Temperature	Thermal injury	Sea surface temperature	10–15	${}^{\circ }$C	[85]
			5–10	${}^{\circ }$C	[86]
Nutrients	Nutrient limitation; Toxic effects	Aqueous nitrate concentration	10	μMNO${}_{3}^{-}$	[87]
Salinity	Osmotic stress	Salinity	24–35	per mille	[87]
			27–33	per mille	[88]
			32	per mille	[86]
Water motion	Drag stress; Diffusion stress	Current velocities	25–1520	mms${}^{-1}$	[88,89]
		Wave heights	6.4	m	[89]

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
