# Peer review of "Biomolecular Composition and Revenue Explained by Interactions between Extrinsic Factors and Endogenous Rhythms of Saccharina latissima"

_marinedrugs, 2019, doi:10.3390/md17020107_

Round 1

Reviewer 1 Report

The paper “Biomolecular composition and revenue explained by interactions between extrinsic factors and endogenous rhythms of Saccharina latissima” reviews the current literature on the cultivation and potential products and revenue from S. latissima biomass. Although this is a good biological review of macroalgae, there needs to be more detail provided around products and revenue and a better description of “triple helix”, since these are in the title of the paper. This paper also needs to be edited by a native English speaker. My specific comments are as follows:

1. Consider editing the title to remove “revenue” if the macroalgae supply chain from cultivation to harvesting to product extraction is not going to be discussed. The paper really just mentions potential products and their cost in the market, but one cannot assume “revenue” (or income to a business after costs) without knowing the costs of production.

2. Line 390, “lights” should be “light” or “wavelengths of light”.

3. Line 386-396, missing some “the”s before words, such as “turbid zone”, “water surface”, “peak”, “deeper zone”, “underwater light”, “Adult”, and others.

4. Line 391-393 and 410-414 are run-on sentences.

5. Line 388, I think you mean “scattered”, not “shattered”.

6. Line 416, change “as well” to “also”.

7. Add space between lines 418 and 419 and 437 and 438 etc.

8. Line 421, degree symbol should go with C.

9. Line 436, “the” Atlantic “Ocean”.

10. Line 490, change “differently” to “in contrast”.

11. Line 493, in “an” algal cell or “the” algal cell. Structure”s”

12. Line 503, “an” offshore or “the” offshore.

13. Line 509, “on”, not “to” S. latissima.

14. Line 523, “the” aquatic.

15. 523-540, this entire section should be edited by a native English speaker. “grazing” not “gazing”, etc.

16. When the author uses “Besides” as a transition, I believe they mean “Also” or “In addition”.

17. Line 646, “years” should not be possessive here. The author could state “years and multiple generations of” adaptations.

18. Line 731, if there are only “a few”, can you list the companies here?

19. 738-739, why is this referred to as a “triple helix”? Is it composed of “seaweed cultivation and biorefinery companies, local authorities and research institutes”, because this is more than three entities? This sentence is confusing.

20. Consider a diagram or figure to describe whatever “triple helix” is.

21. 756-759, this is a very dramatic statement with “seize the control of the lifeblood”. Certainly, the supply chain from growth to harvest to product extraction could be run by one supplier or be a collaborative process.

22. Section 3.3, this section could benefit from a diagram showing the macroalgae supply chain from growth to harvest (is drying required?) to product extraction, displaying which products can be co-purified, and identifying if there are some product extractions that would negate the purification of other products. The entire macroalgae supply chain is important to costs.

23. Line 590-614, it should be noted that protein concentration in algae is often decreased if lipid concentration is increased and vice versa. Lipid production is often a stress response in microalgae – is this also true in macroalgae? Are there existing co-extraction processed where both protein and lipid could be purified?

24. Line 590-614, why no mention of lipid to a biofuel intermediate for a product? Or biomass to biofuel?

25. Line 748-750, the statement “we suggest that phase I biorefinery (i.e. single raw material is utilized for producing one major product and several co-products) is most applicable for business in practice” needs to be justified. How would you envision this? A block flow diagram of this process with potential co-products would illustrate this.

26. Line 769-772, do the authors have some specific industrial evidence they can point to for macroalgae being impacted by climate change? Have any effects been observed in 2018 or 2019 from warmer oceans or increased storms?

27. Line 788-790, this sentence is clumsy.

28. Line 792-794, this statement “Macroalgal culture should be consider at wider geographical scales” needs to be justified (and improved for English grammar). What do the authors mean by “wider geographical scales”? How much coastline is available for macroalgae production? A global map indicating these areas would be illustrative here.

29. 865-869, this sentence is very confusing. No “biorefinery conversion pathways” were discussed in the article.

30. Line 863-864, is this an ash free dry weight? Or a wet weight?

Author Response

We would like to thank the reviewer for the careful and thorough reading of this manuscript and for the thoughtful comments and constructive suggestions, which help to improve the quality of this manuscript.

Please kindly check the uploaded word file of our point-by-point response to the comments.

A comparison between the manuscripts before and after revisions can be found via the link:

https://draftable.com/compare/VPaDxXypXYDE (the one in the right column is the revised version).

Reviewer 2 Report

The manuscript entitled "Biomolecular composition and revenue explained by interactions between extrinsic factors and endogenous rhythms of saccharina latissimi" by Zhang X et al describes the important aspects of seaweed growth, revenue, seasonal conditions on yield etc.as a comprehensive review.

1) In section 2.1, please explain how AND operator relates to S. latissimi?

2) How exactly the spatial variations interrelates S. latissima growth and revenue?

3) In section 2.3, what exactly valorization method refers to, please also explain what kind of valorization that influences the revenue analysis? similarly in section 4.2.2, how valorization and what type of valorization that affects the farming of seaweed.

overall, the review sounds good with factors influenced the growth and revenue of seaweed.

recommended for minor revisions only.

Author Response

(The authors gave the same response as above.)

Reviewer 3 Report

Nice review, everything is well explained and structured, covered all relevant references. I believe the whole coverage is appropriate for publication in "Marine Drug".

Author Response

We would like to thank the reviewer for the careful reading of this manuscript and the positive feedback.

Please kindly check the uploaded word file of our response to the comments.

A comparison between the manuscripts before and after revisions can be found via the link:

https://draftable.com/compare/VPaDxXypXYDE (the one in the right column is the revised version).

Round 2

Reviewer 1 Report

The authors addressed all of my comments sufficiently. With the new text additions, there are still some grammar issues which should be checked. For example, "Economic feasibility of the supply chain from seaweed nursery, deployment, growth to harvest and product extraction is obtained under the circumstances; first, the revenue of the harvested seaweed exceeds the cost of biomass production; and second, the costs of the feedstock and the high-end technologies used to extract high-value biomolecules are exceeded by the revenue from the sales of final output extract products [148]." should be edited for sentence structure such as, "Economic feasibility of the supply chain from a seaweed nursery, to deployment, to growth to harvest and to product extraction is obtained under the following circumstances: First, the revenue of the harvested seaweed exceeds the cost of biomass production; and second, the costs of the feedstock and the high-end technologies used to extract high-value biomolecules are exceeded by the revenue from the sales of final output extract products [148]." Also, "“Throughout the years and multiple generations’ adaption,” is still grammatically clumsy and should be changed to "Over time and multiple generations of adaptation" for example.

All in all, this is an interesting review and will be a significant contribution to the field.

Author Response

We are very grateful for the comments provided by reviewer 1. The comments are encouraging and the reviewer appears to share our judgement that this review is important.

We have made suggested changes to the sentences mentioned by the reviewer. In addition, we have made a grammar check for the entire manuscript.

Please use the link below to compare the revised document (right column) with the previous version to track the changes we have made:

https://draftable.com/compare/WXNgwkKIYCQd 

Sincerely,

Marianne Thomsen & Xueqian Zhang

This manuscript is a resubmission of an earlier submission. The following is a list of the peer review reports and author responses from that submission.